# A Bibliometric Review of the Evolution of Blockchain Technologies

**DOI:** 10.3390/s23063167

**Published:** 2023-03-16

**Authors:** Sergi López-Sorribes, Josep Rius-Torrentó, Francesc Solsona-Tehàs

**Affiliations:** 1Department of Computer Science, University of Lleida, 25001 Lleida, Spain; 2Department of Economics, University of Lleida, 25001 Lleida, Spain

**Keywords:** blockchain, bibliometric analysis, scientific collaboration, topic analysis

## Abstract

Bitcoin was created in 2008 as the first decentralized cryptocurrency, providing an innovative data management technology, which was later named blockchain. It ensured data validation without intervention from intermediaries. During its early stages, it was conceived as a financial technology by most researchers. It was not until 2015, when the Ethereum cryptocurrency was officially launched worldwide, along with its revolutionary technology called smart contracts, that researchers began to change their perception of the technology and look for uses outside the financial world. This paper analyzes the literature since 2016, one year after Ethereum, analyzing the evolution of interest in the technology to date. For this purpose, a total of 56,864 documents created between 2016 and 2022 from four major publishers were analyzed, providing answers to the following questions. Q1: How has interest in blockchain technology increased? Q2: What have been the major blockchain research interests? Q3: What have been the most outstanding works of the scientific community? The paper clearly exposes the evolution of blockchain technology, making it clear that, as the years go by, it is becoming a complementary technology instead of the main focus of studies. Finally, we highlight the most popular and recurrent topics discussed in the literature over the analyzed period of time.

## 1. Introduction

Bitcoin was introduced by Nakamoto in 2008 [1], as the first decentralized currency that allowed for currency exchange between peers without any central intermediary. Among other important features, the major ledger is managed in a distributed way between all members of the network, who do not necessarily trust each other. This is an attempt to solve the famous Byzantine generals problem [2]. In this scenario, the generals have encircled Byzantium, but they must decide when to assault as a group. They will win if all generals assault simultaneously. Because any letters they transmit or receive could have been intercepted or deceptively sent by Byzantium’s defenders, the generals have no secure communication channels. This is about providing people with a way to communicate safely and securely in an unpredictable world. In the actual world, most transactions occur between strangers who do not know or trust one another.

The Byzantine generals problem can be exemplified by money: how could a society build a monetary system that all members can trust and agree on? Bitcoin was the first realized solution to the Byzantine generals problem [2]. To achieve this, Bitcoin uses what is called a consensus mechanism, a fault-tolerant mechanism in which the nodes reach an agreement in the creation of a valid block. There are various consensus mechanisms, but Bitcoin uses the one called Proof-Of-Work (POW) [3]. POW is a form of cryptographic proof in which one party proves to the others that a certain amount of a specific computational effort has been expended. The key point of POW is that this computational effort can be proved by the other nodes with minimal effort.

The popularity of Bitcoin sparked scientific interest, encouraging the community to create many other cryptocurrencies. This is the case for Litecoin [4], a decentralized currency based on Bitcoin with similar parameters. The main difference is in the block (Block: Blocks are data structures on which each transaction is recorded; the blockchain is made of chronologically linked blocks) generation time and in the maximum amount of coins that can be generated in the network. A Litecoin block is generated every 2.5 min, four times faster than Bitcoin. This means that Litecoin can process more transactions per second, making it more suitable for small transactions. Regarding the maximum number of coins that can be generated by a mining process (Mining: Is the process that makes each node to reach a consensus and generate a new block with their respective reward to the first to generate the block), Bitcoin has an inherent maximum of 21 million coins, while Litecoin’s maximum is 84 million.

Another example of decentralized currency is Primecoin [3]. Unlike Bitcoin, which uses SHA-256, Primecoin was the first cryptocurrency with a non-hashcash consensus mechanism.

PrimerCoin’s consensus mechanism is based on searching and verifying strings of prime numbers, such as the bit-twin and Cunningham string. Its characteristics include a block generation time of 1 min, 10 times faster than Bitcoin, and an estimated block verification time that is 8–10 times faster than that of Bitcoin.

Other scientists, instead of developing a new structure from scratch when creating their cryptocurrency, decided to use Bitcoin’s structure and provide more functionality.

This is the case, for example, for ColoredCoind [1], which is built on the top level of Bitcoin blockchain and converts the normal fungible (Fungible: Bitcoin theoretically is considered fungible, which means that one BTC owned by A can be exchanged for one BTC owned by B, since they have the same value.) to non-fungible Bitcoins, coloring a set of coins to differentiate them from the others. This allows for these coins to have special proprieties and they can represent different assets, allowing for the exchange of real things in a decentralized way.

At this point, four years after the appearance of Bitcoin, the bulk of the investigations were focused on the financial sector. This created the wrong impression that blockchain, as a technology, is only applicable to the financial sector. This perception started to change in 2013 with the emergence of Ethereum [5], a new decentralized currency. This provided an important feature. Ethereum was the first cryptocurrency presented as a Turing complete system, allowing for participants to create parts of the code that are linked to the blockchain and invoice them by establishing a transaction on this chain.

These parts of the code are known as smart contracts. This new feature was very different from the existing cryptocurrency and, for this reason, Ethereum became the reference system since its launch in 2015, changing the perception that blockchain technology was only for financial purposes.

This work aims to analyze the evolution of blockchain research since the launch of Ethereum.

To achieve this, we performed a bibliometric analysis to obtain an objective research focus. A bibliometric analysis is a quantitative analysis of scientific publications such as books or articles that can help to understand the status of a technology, as well as any hotspots. The interest of the scientific community in blockchain technology between 2016 and 2022 was analyzed in an attempt to answer the following questions:Question 1: How has the interest in blockchain technology increased?Question 2: What have been the major blockchain research interests?Question 3: What have been the most outstanding works of the scientific community?

There are other works with similar objectives, such as [6,7,8]. However, it was noted that all of them used a more generic approach. Furthermore, the study lacks information about the evolution over recent years. For this reason, we believe that this work will provide another perspective and extend the work that has been carried out to date.

The rest of the paper is structured as follows. Section 2, Methods, explains how the information was obtained and processed. The results obtained by applying these methods are shown in Section 3, Results, and discussed in Section 4, Discussion, along with the limitations that were found. The conclusions of this paper and future work can be found in Section 5, Conclusions.

## 2. Methods

The corpus analyzed in this study was obtained by an application developed by our group [9]. The dataset was compiled from four reputable publishers: IEEE, SPRINGER, ELSEVIER and ACM. It consists of scientific documents published between the years 2016 and 2022, both inclusive, that contained the keyword blockchain in either their keywords or title. A data visualization tool is also made available for further analysis.

Next, the methodology employed by our application to process and cleanse the corpus in accordance with each specific inquiry is outlined.

### 2.1. Question 1: How the Interest in Blockchain Technology Increased?

We tackled the first question by processing all the documents obtained in the indicated period (years 2016–2022). Then, we used the number of documents found in 2016 as a basis to compute the annual and total variation each year. In this way, we can understand how the research community varied their efforts to investigate blockchain technology.

### 2.2. Question 2: What Areas Have Been Investigated the Most?

To find the most interesting research areas for the scientist community, we analyzed all the document titles for the analyzed period of time, applying the following filters and transformations:Remove all numbers.Transform all characters to lower-case.Remove all stop words.Remove non-meaningful words for analysis: Blockchain, based, research, analysis, use, data, chapter, index, block, review, approach, survey, system, technology, towards, etc.Transform all words to their respective root word: chains-chain.Group words or groups of words with a similar meaning under the same word, e.g., IoT: Internet of Things, IoT devices; IA: machine learning, deep learning.

Once these filters and transformations were applied, a word count of the blockchain-related title words was made. The most 10 popular terms per year were also selected.

### 2.3. Question 3: What Have Been the Most Outstanding Works of the Scientific Community?

To find the most relevant documents generated per year, we used the number of citations for each document. Thus, the top three most popular articles related to blockchain technology per year were obtained.

## 3. Results

The corpus was made of 56,864 documents published between 2016 and 2022. 26,426 of them were from the SPRINGER publisher, 10,613 from ELSEVIER 15,475 from IEEE and 4350 from ACM.

Table 1 shows the number of papers published in each editorial per year.

### 3.1. Question 1: How Has Interest in Blockchain Technology Increased?

The publication of scientific documents with a direct or indirect mention of the word blockchain increased over the years, reaching increments of 296% in the early years. This increment reflects a growing interest in blockchain technologies from the scientific community.

Since the first year of the analysis, for which 310 documents were found, until the last year, for which 19,305 documents were found, the number of published papers discussing this technology increased by 62 times. Figure 1 shows this evolution.

### 3.2. Question 2: What Have Been the Major Blockchain Research Interests?

Figure 2 shows the evolution of the top 10 most-used words per year in scientific publications from 2016 to 2022.

A total of 16 words appeared in this top 10 list per year at least once. These were as follows:Bitcoin: The first known cryptocurrency to use blockchain.Security: Technical aspects and resources providing secure actions over the Internet.Contract: Blockchain applications responsible for invoking transactions.Distributed: Intrinsic feature of blockchain. The processes are carried out between various entities in the chain.Future: Theories and speculations about future trends.Management: Handle control, processing and data.Privacy: Keep data away from unauthorized third parties.Financial: Related to money or financial transactions.IoT: Internet of Things: Devices equipped with sensors, processing abilities, software, and other technologies that connect and exchange data with other devices and systems over the Internet.Application: Blockchain-based applications.Digital: Resources that are implemented or saved digitally.Decentralized: Processes carried out without a central entity.Energy: Using blockchain for peer-to-peer energy management.Model: Concepts used to learn, understand, or simulate the subject the model represents.Framework: A platform system on which applications may be run.Supply: Refers to the sequence of events that transpire from the point at which a customer places an order to the delivery and payment of the product or service.

Three of these words (security, IoT and application) appeared in the studied period. Their position grew each year until it appeared at the top position in the last year of the analysis.

Significant growth was also observed in the energy (2018) and framework (2019) terms, occupying the sixth and eighth positions, respectively, in the last year of analysis.

### 3.3. Question 3: What Have Been the Most Outstanding Works of the Scientific Community?

The top three most outstanding papers for each year were compiled from the most-referenced documents that contain the word blockchain, even if blockchain technology was the main topic or it was proposed as an improvement in or solution to another topic.

For each document, the following points were highlighted:A summary of the document.The role of blockchain in the paper: is it the main theme, or is it an enhancement to another theme, or is it only mentioned?The kind of paper: original, perspective, case study, review or Special Issue?

Table 2 shows all the documents compiled for each year with the number of citations and the kind of paper, indicating whether blockchain was the main topic. The following sections contain excerpts from this table, filtered by the year of the section.

#### 3.3.1. 2016

Two of the top three most highly cited articles in 2016 belong to advancements in blockchain technology, while the remaining publication is a comprehensive review (see Table 3).

The most-referenced paper in 2016 [10] was a review in which the blockchain technology and smarts contracts were detailed and linked to IoT technology. It highlights the benefits of using these technologies together. It concludes that synergy can be strong, allowing for the creation of automated workflows in ways never seen before, with cryptographic verification and important reductions in the processing time.

In the second most-referenced paper [11], the authors showed their concerns regarding the transparency of the economic transactions related to blockchain technologies and the traceability of third parties. To solve this issue, they proposed the use of Hawk, a custom framework that allows for programmers to implement smart contracts in an intuitive way, without having to worry about cryptography, which is taken care of by the compiler.

The third cited publication is a seminal work [12] that highlights the imperative for innovation in electronic medical records (EMR). The authors present a proposal for a public blockchain-based EMR system, which is designed to provide patients with comprehensive, tamper-proof records and convenient access to their medical information, while preserving confidentiality and security.

#### 3.3.2. 2017

This year, the top three most-cited documents were two reviews and one case study (see Table 4).

The top paper in 2017 was a review [13]. It analyzed the blockchain technologies, discussing their architecture, the advantages of the different consensus mechanisms and some barriers that blockchain must surpass to be accepted. As a concluding remark, the article proposes investigating blockchain-based applications to improve the technology.

In the second place was another review [14] of the so-called 4.0 industries and how their technology can improve business logistics. In this article, the blockchain is not the main topic, but features as a related technology in Industry 4.0. The document talks about the use of blockchain as a means to improve the supply chain. The authors concluded that Industry 4.0 is a marketing label, which tries to group all the new technologies together.

In the third position, we found [15], a case study that showed how blockchain can work together with IoT. These technologies were applied to a smart home scenario. The authors reduced the cost of implementing blockchain with IoT technologies by means of a reduced blockchain protocol in terms of security and privacy. They showed that the proposed reduced-blockchain-based smart home framework provided integrity, availability, and confidentiality.

#### 3.3.3. 2018

Table 5 shows the top three articles for 2018. As for the previous year, we found one original work and two reviews.

The most-cited paper [16] introduces Fabric, an open-source system for deploying authorized blockchains. It is presented as an innovative alternative to existing solutions from the time. Its strengths include modularity in the choice of consensus system, the ability to run distributed applications written in standard programming languages, and the lack of dependence on a native cryptocurrency system. Additionally, the authors demonstrated the novel creation/design of the blockchain and how the new design addresses non-determinism, resource exhaustion, and performance attacks. This is supported by comparative tests with other cryptocurrency systems based on Bitcoin, demonstrating its significant superiority.

The second [17] of 2018 is a review of IoT. The document highlight the security problems with IoT technology and compiles the related work with the objective of proposing a solution based on blockchain technology.

The third most-cited paper [18] for that year was another review about the applicability of blockchain in IoT. It explains how these two technologies match each other and highlights the contribution of smart contracts to their synergy, discussing the different consent mechanisms, depending on blockchain type. An interesting remark of the authors regards the different barriers that must be overcome by the blockchain to improve its deployment in IoT, and affirm that blockchain could revolutionize this field.

#### 3.3.4. 2019

On Table 6, we can see the top three most-cited documents of 2019. In this case, all of them are reviews.

The most-cited paper [19] of 2019 was a compressive review. The authors present an analysis of 282 works on digital transformation (DT) (Digital transformation: The process of integrating digital technologies into business processes with the aim of improving productivity and/or customer experience.) to explore their role in the creation and reinforcement of industry and society. The review employs an analytical framework composed of eight blocks: disruptions, strategic responses, use of digital transformation, changes in value-creation paths, structural changes, organizational barriers, negative impacts, and positive impacts. The analysis shows that the growth in DTs has had a considerable impact on the complexity of companies and society. The review highlights that DTs have resulted in significant changes in the way that businesses operate and brought about new opportunities for value creation. However, they also presented a range of challenges, including organizational barriers and negative impacts.

The second [20] is a systematic review. The authors of this publication present a comprehensive evaluation of the potential impact of blockchain technology in the energy sector. They analyze various use cases and identify the potential benefits, challenges, and limitations of each application, including P2P energy trading, IoT, decentralized marketplaces, electric vehicle charging and e-mobility. They conclude that while blockchains offer significant benefits, such as transparency and tamper-proof transactions, these systems must overcome various challenges and barriers to achieve widespread adoption.

The last paper [21] of the top three is a review of the current state of blockchain technology and its applications across multiple domains, such as supply chain, business, healthcare, IoT, privacy and data management. The authors conclude that while blockchain applications are widely deployed, many issues still need to be addressed, and traditional databases may be a better option for some scenarios.

#### 3.3.5. 2020

The top three articles for this year, were one Special Issue and two reviews. It is important to highlight that two of them were about COVID-19 (see Table 7). Please note that this is an atypical year due to the COVID-19 pandemic and this fact contributed to the top three articles that were found.

The top paper [22] for this year is a Special Issue where the impact of the COVID-19 pandemic on society and the global economy are discussed. It addresses some of the pandemic-related issues affecting various industrial sectors, consumer behavior, business and ethical issues. The Special Issue includes 13 articles covering a wide variety of topics related to the pandemic and its effects on different aspects of society.

The second most-cited paper [23] of the year was a blockchain review. It details the known vulnerabilities in general blockchains and smart contracts and explain the different attacks according to each vulnerability. The article concluded by presenting ideas from different research areas to resolve these vulnerabilities.

The third [24] was a review about the status of COVID-19 at that moment. One part of the paper explained the different technologies being used. Between these, blockchain was very commonly used, including for the following:Facilitating increased testing and reporting;Recording the details of the COVID-19 patients;Managing the lockdown implementation;Preventing the circulation of fake news;Enabling an incentive-based volunteer participation platform;Enabling a secure donation platform for supporters;Limiting supply chain disruptions.

#### 3.3.6. 2021

The top three most-cited articles in 2021 were fully composed of papers that expose the authors’ perspective of new technologies, such as IA, blockchain, and robots (see Table 8).

The first article [26] was a compilation of the opinion of various experts regarding the impact of artificial intelligence technology and its challenges in the fields of business and management, government, public sector, science and technology.

The second paper [25] was a literature review about digital transformation. The authors show different strategies and their requirements for the growth of digital companies. Finally, they provided a research agenda and a guide for future research on digital transformation.

The last paper [27] of the top three was a compilation of the collective insights of several leading experts on issues relating to digital and social media marketing. These perspectives offer a detailed narrative of the key points, as well as a more specific look at key aspects such as artificial intelligence, augmented reality marketing, digital content management, B2B marketing, mobile marketing and advertising.

#### 3.3.7. 2022

Table 9 shows the top three most-cited documents of 2022. One of them is a perspective view regarding the incoming future, and the other two are reviews.

The top paper [28] for 2022 was a perspective about Industry 5.0 (Industry 5.0: The term first appeared in 2021, when the European Union aimed to refocus technological development so that the emphasis was not only on increasing efficiency but also on creating a positive social impact.). The authors of the article begin their speech by addressing the notion of Industry 5.0, presenting a complete description of the concepts and definitions that compose this term. Then, they delve into an in-depth examination of the potential applications of Industry 5.0, including but not limited to smart healthcare systems, blockchain technology, cloud-based manufacturing, supply chain optimization, and advanced production methods. Finally, the authors highlight several important research challenges and open problems that require further investigation before the full potential of Industry 5.0 can be realized.

In second place is a review [29] of the existent research regarding the integration of Internet of Things (IoT) and Unmanned Aerial Vehicles (UAVs) (UAVs: also known as drones, these comprise all aircraft that fly without a human crew.) in precision agriculture. The authors posit that these technologies could play a crucial role in transforming traditional agriculture practices into a new era of precision agriculture, also known as Agri-Food 4.0. The authors emphasize the importance of future research, focusing on the key agronomic aspects of smart farming, including precise irrigation with salt minimization, efficient and rational fertilization and pesticide use, effective weed management, efficient crop disease management, the implementation of 3D modeling algorithms for crop growth monitoring, use of blockchain technology for food traceability and accurate yield prediction in agriculture.

In third position was [30], a review of the convergence of the 6G and the Internet of Things (IoT). The authors’ goal was to investigate the emerging opportunities brought by the integration of 6G technologies into IoT networks and applications. They highlight the 6G Fundamental technologies that are anticipated to increase the capabilities of future IoT networks, including edge intelligence, space-air-ground-submarine communications and blockchain. The authors conclude that the 6G has attracted substantial attention in recent years due to its advantageous features. They predict that the 6G will bring about a transformation in the current IoT network infrastructure and lead to improvements in the quality of service and the user experience in future applications.

## 4. Discussion

This section use the data found in the previous Section 3, Results, to examine the questions exposed in Section 1, Introduction, providing a discussion about the questions that were raised and addressing the limitations.

### 4.1. Question 1: How Has Interest in Blockchain Technology Increased?

First, we want to talk about the growing interest in blockchain technology from the scientific community. As we can see in Table 1, Figure 1 and Section 3.1, the number of papers mentioning this technology—as the main topic or as a supporting topic—experienced a linear growth during the analyzed period (years 2016–2022).

Significant results were obtained for the first question, Q1: How has interest in blockchain technology increased? Since the launch of Ethereum in 2015, the annual generation of scientific papers referencing blockchain linearly increased. More specifically, comparing the last year (2022) to the first one (2016), the obtained results show that the number of papers were multiplied by a factor of 62.

### 4.2. Question 2: What Have Been the Major Blockchain Research Interests?

As mentioned in the introduction, during the first years of the Bitcoin lifetime, the feeling in the community was that blockchain technology was only applicable to the financial sector, but this belief completely changed following the release of Ethereum. If we take a look at Figure 2, we can see as the word bitcoin went from the top position in the years 2016 and 2017 to the third one in 2018, and finally disappeared from this ranking after 2018. Moreover, references to any financial aspect disappeared from the top ten positions after 2018. This fact confirms that the general perception that the application of blockchain to financial areas gradually lost interest, with Bitcoin gaining in popularity and application in other research areas, such as IoT or energy.

This change in research trends may be due the inheritance characteristic of physical decentralization, weak security protocols that need to be improved, the need to expose the data to Internet without restrictions, or any other issue related to IoT. The use of these two technologies together, blockchain and IoT, increased in popularity from the start of the study. IoT was ranked in the top nine in 2016, but soon started to share the top two positions with the term security. This term seems to be strongly related to the blockchain technology, since it never moved from the top two positions during the analyzed period.

Three other concepts have received limited recognition compared to the well-established terms of security and IoT. However, these concepts consistently remained within the top 10 and experienced an upward trend in popularity over time. One of these concepts is privacy, which is widely regarded as a key attribute within the blockchain paradigm. The other concept is application, which demonstrated a significant growth, progressing from its initial ranking of 8th in the first year to 4th in the most recently analyzed year. This increase in popularity highlights the growing interest and investment of the scientific community in exploring the potential applications of these technologies across various domains. The final term of significance is management, which progressed from being ranked 6th in the second year to 5th in the most recent year. These results are consistent with the expectations, as blockchain possesses numerous desirable properties that are beneficial for management. For instance, blockchain enables decentralized data management with or without a third party, eliminates the need for mutual trust among participants, facilitates autonomous management through smart contracts and provides a traceable record of all actions taken on the chain, among others.

Figure 2 shows the overall terms regarding popularity over the years. For example, the terms contract and distributed were very relevant the first years of the analysis but in recent years, they are not even present. This may be due the growing interest in different areas, such as framework and energy. The framework term, for example, appears in the first position for first time in 2019. This term includes methodologies or programs to speed up blockchain management, and the development of new projects based on this technology. energy also showed a growing interest in relation to blockchain since 2018. An explanation for this could be found regarding the importance of the two main topics. The first one is the interest in the energy consumption of blockchain systems and the second one is the use of blockchain technology in projects, with the aim of sharing energy in a decentralized way.

At this point, using all information regarding the evolution of the most popular terms related to blockchain, we can answer question two. We can observe that, over all the analyzed years, four popular terms were always present in the ranking. They were not always in the top positions, but were always present: security, IoT, application and privacy. The consistent presence of these terms within the top 10 rankings implies a robust synergistic relationship. This is further supported by the proliferation of scientific works that endorse the integration of these terms to enhance the state of the art [31,32,33,34].

There were also two emergent terms with an important and constant increase in popularity in recent years. These were: energy and framework. The terms energy and framework represent a latent potential, a research challenge in its first steps of exploration, which is becoming an important scientific interest. Finally, to conclude the discussion, we address the last question. Table 2 shows the three most-referenced articles per year that refer to blockchain technology. It can be observed that, in the first years of study, most papers were focused on blockchain technology as the main topic [10,11,12,13,15,17,18,20,21,23,35].

Recently, the main focus changed to more specific blockchain technology solutions in various fields, such as security, data sharing, artificial intelligence, IoT and energy [14,19,22,24,25,26,27,28,29,30].

### 4.3. Question 3: What Have Been the Most Outstanding Works of the Scientific Community?

At early stages, blockchain-like technologies were in an immature state, with unknown potential. This explains why most of the studies that refer to blockchain aimed to clarify blockchain concepts and operation or tried to implement them in a real scenario. Over the years, the foundations for blockchain technology have been laid, causing many authors to see the technology as a reinforcement to many other areas that were not covered before.

### 4.4. Limitations of the Review

The present study acknowledges two significant limitations. First, the use of a more extensive pool of publishers is recommended; however, we opted to concentrate on recognized and representative scientific publishers. Second, our analysis is restricted to scientific papers published by registered publishers to trace the progress of scientific interest throughout the years. As a result, it excludes public or business interests from the scope of analysis.

## 5. Conclusions

In this work, we conducted a bibliographic review of all the research that contained the term blockchain published in the main scientific editorials since 2016—a year before Ethereum’s launch—until 2022, with the main objective of answering three questions. How has interest in technology increased? What have been the major blockchain research interests? What have been the most outstanding works of the scientific community? To do this, the four main and more prestigious editorials—IEEE, SPRINGER, ELSEVIER and ACM—were chosen. A corpus with 56,864 papers was obtained.

We analyzed the corpus, considering the year of publication and the interest in the blockchain technologies. In 2016, 310 papers were published, and 19,305 were published in 2022, with an ascendant linear growth being shown throughout the entire period.

Then, we counted the blockchain words in each document title. Non-significant words were discarded. This allowed for us to generate a top ten most-used terms per year. Using this ranking, we observed how the financial concept of blockchain loses strength over the years. This place was filled by other concepts, such as security, IoT, management, application and privacy. The emerging concepts were consolidated into the top positions. It is also important to highlight that the terms Energy and Framework showed a constant growth since their first appearance.

To cover the subject in more depth, the top three most-referenced manuscripts per year were analyzed. This analysis showed that, each year, the core of the documents stopped focusing on the blockchain itself, and it started to become a complementary technology for many other topics and applications. This trend proves that the scientific community has become more aware of the capabilities, applications and new synergies. This fact has led to the use of blockchain as a solution to different problems, or as an alternative functionality in other technologies.

As future work, it would be worth examining the progression of interest in the emerging scientific fields uncovered in this research, as well as conducting comparable analyses with complementary publishers to validate the findings. In addition, assessing the perception and interest in blockchain in the public and business domains using a comparable analysis would be a compelling topic for future research.

## Figures and Tables

**Figure 1 sensors-23-03167-f001:**
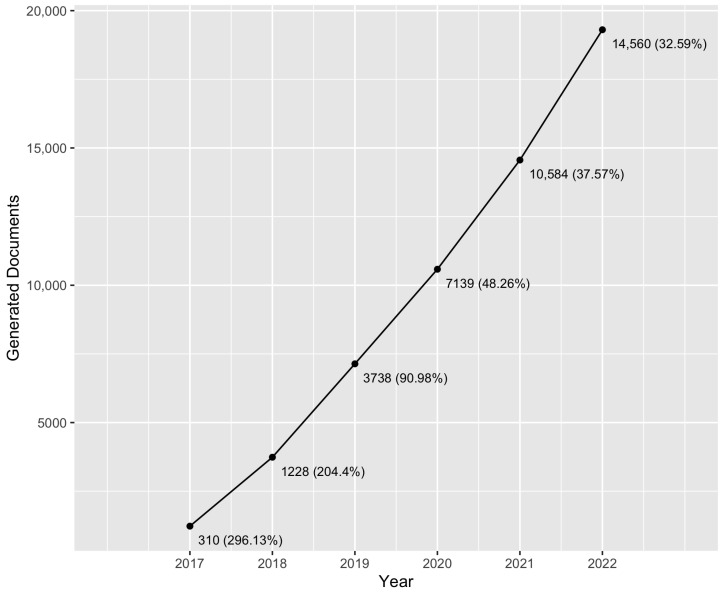
Collected documents per year.

**Figure 2 sensors-23-03167-f002:**
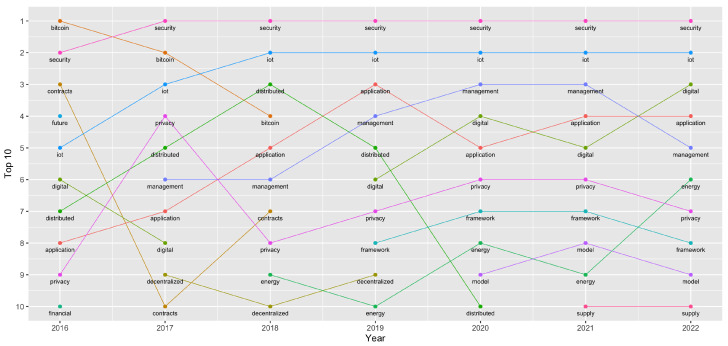
Top 10 most-used words and their evolution.

**Table 1 sensors-23-03167-t001:** Editorials’ blockchain document creation.

Platform	2016	2017	2018	2019	2020	2021	2022
ACM	64	188	447	738	801	1016	1096
IEEE	75	325	1434	2507	3166	3768	4200
SPRINGER	139	598	1367	2854	4753	6860	9855
ELSEVIER	32	117	490	1040	1864	2916	4154
ALL	310	1228	3738	7139	10,584	14,560	19,305

**Table 2 sensors-23-03167-t002:** Top 3 citations by year.

Year	Title	Quotes	M.topic?	Type
2016	Blockchains and Smart Contracts for the Internet of Things [10]	2331	YES	Review
2016	Hawk: The Blockchain Model of Cryptography and Privacy-Preserving Smart Contracts [11]	1077	YES	Original
2016	MedRec: Using Blockchain for Medical Data Access and Permission Management [12]	1000	YES	Original
2017	An Overview of Blockchain Technology: Architecture, Consensus, and Future Trends [13]	1601	YES	Review
2017	Industry 4.0 and the current status as well as future prospects on logistics [14]	922	NO	Review
2017	Blockchain for IoT security and privacy: The case study of a smart home [15]	807	YES	Study case
2018	Hyperledger fabric: a distributed operating system for permissioned blockchains [16]	1671	YES	Original
2018	IoT security: Review, blockchain solutions, and open challenges [17]	1443	YES	Review
2018	On blockchain and its integration with IoT. Challenges and opportunities [18]	999	YES	Review
2019	Understanding digital transformation: A review and a research agenda [19]	1126	NO	Review
2019	Blockchain technology in the energy sector: A systematic review of challenges and opportunities [20]	1123	YES	Review
2019	A systematic literature review of blockchain-based applications: Current status, classification and open issues [21]	968	YES	Review
2020	Effects of COVID-19 on business and research [22]	757	NO	Special Issue
2020	A survey on the security of blockchain systems [23]	689	YES	Review
2020	A Comprehensive Review of the COVID-19 Pandemic and the Role of IoT, Drones, AI, Blockchain, and 5G in Managing its Impact [24]	602	NO	Review
2021	Digital transformation: A multidisciplinary reflection and research agenda [25]	572	NO	Review
2021	Artificial Intelligence (AI): Multidisciplinary perspectives on emerging challenges, opportunities, and agenda for research, practice and policy [26]	556	NO	Perspective
2021	Setting the future of digital and social media marketing research: Perspectives and research proposition [27]	363	NO	Perspective
2022	Industry 5.0: A survey on enabling technologies and potential applications [28]	214	NO	Perspective
2022	Internet of Things (IoT) and Agricultural Unmanned Aerial Vehicles (UAVs) in smart farming: A comprehensive review [29]	158	NO	Review
2022	6G Internet of Things: A Comprehensive Survey [30]	98	NO	review

**Table 3 sensors-23-03167-t003:** Top 3 citations 2016.

Quotes	Title	M.topic?	Type
2331	Blockchains and Smart Contracts for the Internet of Things [10]	YES	Review
1077	Hawk: The Blockchain Model of Cryptography and Privacy-Preserving Smart Contracts [11]	YES	Original
1000	MedRec: Using Blockchain for Medical Data Access and Permission Management [12]	YES	Original

**Table 4 sensors-23-03167-t004:** Top 3 citations 2017.

Quotes	Title	M.topic?	Type
1601	An Overview of Blockchain Technology: Architecture, Consensus, and Future Trends [13]	YES	Review
922	Industry 4.0 and the current status as well as future prospects on logistics [14]	NO	Review
807	Blockchain for IoT security and privacy: The case study of a smart home [15]	YES	Study Case

**Table 5 sensors-23-03167-t005:** Top 3 citations 2018.

Quotes	Title	M.topic?	Type
1671	Hyperledger fabric: a distributed operating system for permissioned blockchains [16]	YES	Original
1443	IoT security: Review, blockchain solutions, and open challenges [17]	YES	Review
999	On blockchain and its integration with IoT. Challenges and opportunities [18]	YES	Review

**Table 6 sensors-23-03167-t006:** Top 3 citations 2019.

Quotes	Title	M.topic?	Type
1126	Understanding digital transformation: A review and a research agenda [19]	NO	Review
1123	Blockchain technology in the energy sector: A systematic review of challenges and opportunities [20]	YES	Review
968	A systematic literature review of blockchain-based applications: Current status, classification and open issues [21]	YES	Review

**Table 7 sensors-23-03167-t007:** Top 3 citations 2020.

Quotes	Title	M.topic?	Type
757	Effects of COVID-19 on business and research [22]	NO	Special Issue
689	A survey on the security of blockchain systems [23]	YES	Review
602	A Comprehensive Review of the COVID-19 Pandemic and the Role of IoT, Drones, AI, Blockchain, and 5G in Managing its Impact [24]	NO	Review

**Table 8 sensors-23-03167-t008:** Top 3 citations 2021.

Quotes	Title	M.topic?	Type
572	Digital transformation: A multidisciplinary reflection and research agenda [25]	NO	Review
556	Artificial Intelligence (AI): Multidisciplinary perspectives on emerging challenges, opportunities, and agenda for research, practice and policy [26]	NO	Prospective
363	Setting the future of digital and social media marketing research: Perspectives and research proposition [27]	NO	Perspective

**Table 9 sensors-23-03167-t009:** Top 3 citations 2022.

Quotes	Title	M.topic?	Type
214	Industry 5.0: A survey on enabling technologies and potential applications [28]	NO	Perspective
158	Internet of Things (IoT) and Agricultural Unmanned Aerial Vehicles (UAVs) in smart farming: A comprehensive review [29]	NO	Review
98	6G Internet of Things: A Comprehensive Survey [30]	No	Review

## Data Availability

The processed dataset is available at: https://github.com/slopez1/ScientificAnalyzer (accessed on 3 March 2023).

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
