# Peer review of "A Bibliometric Review of the Evolution of Blockchain Technologies"

_sensors, 2023, doi:10.3390/s23063167_

Round 1

Reviewer 1 Report

English language correction is suggested in the entire manuscript.

The authors made a statement "Finally, the most stable and emerging fields over time are found." in the abstract, but I think this will be so early to say it for Blockchain technology. Maybe authors should give their context/opinion with the statement.

The authors say that they will try to find the answer to three questions but have not mentioned why it is significant and why it is required to get these answers.

The paper seems to be a status report of blockchain-related literature. The answers to the three asked questions are just statistical and no real efforts were made to actually find out the answers. For example, the magnitude of published papers in terms of numbers is used to give all the answers. It will be interested to know why it is important to know the answers to these three questions.

Reviewer 2 Report

This paper presents a review of published papers in the field of blockchain, providing three major findings to address the proposed research questions (Q1-Q3). The work offers valuable insights into the trends and evolution of blockchain over the past few years. While academic papers on blockchain may lag behind industry developments, this work partially captures trends up until 2021.

It's worth noting that the dataset excludes ACM, a major publisher of blockchain research, including flagship conferences such as WWW, CCS, CSUR, TC, and others. I strongly recommend including this publisher to enhance the robustness and completeness of the study.

Regarding the Discussion section, I suggest summarizing the discussion into several independent sub-topics to improve clarity.

The blockchain community has recently ventured into more innovative fields, such as NFTs, Web3, Metaverse, and DeFi. While this paper cannot reflect these changes in real time, I am eager to see further investigations analysing recent years in these areas.

Minors:

Typos in the title: “Tecnologies”  Technologies

Reviewer 3 Report

Sensors

sensors-2264007

Comments

A Bibliometric Review of the Evolution of Blockchain Technologies

The paper is based a novel contribution to research and scientific community using Blockchain technology inspired healthcare , but the following cages are required with minor revisions:

1. A though proofreading is required in the overall paper

4. The author need to add the following references in the revised version of the paper:

Ali, A.; Almaiah, M.A.; Hajjej, F.; Pasha, M.F.; Fang, O.H.; Khan, R.; Teo, J.; Zakarya, M. An Industrial IoT-Based Blockchain-Enabled Secure Searchable Encryption Approach for Healthcare Systems Using Neural Network. Sensors 202222, 572. https://doi.org/10.3390/s22020572

Round 2

Reviewer 1 Report

The paper can be accepted in the present form.

Reviewer 2 Report

The authors have made efforts to address the issues raised by the reviewers, particularly by including one more reputational ACM corpus. While some emerging technologies in the blockchain field have been omitted due to academic delays, this work remains acceptable.